# Neoadjuvant cobimetinib and atezolizumab with or without vemurafenib for high-risk operable Stage III melanoma: the Phase II NeoACTIVATE trial

Tina J. Hieken[1], Garth D. Nelson[2], Thomas J. Flotte[3], Eric P. Grewal[4], Jun Chen[5], Robert R. McWilliams[4], Lisa A. Kottschade[4], Lu Yang[5], Evidio Domingo-Musibay[6], Roxana S. Dronca[7], Yiyi Yan[7], Svetomir N. Markovic[4,8], Anastasios Dimou[4], Heather N. Montane[4], Courtney L. Erskine[8], Mara A. Piltin[1], Daniel L. Price[9], Samir S. Khariwala[10], Jane Hui[11], Carrie A. Strand[2], Susan M. Harrington[8,12], Vera J. Suman[2], Haidong Dong[8,12] & Matthew S. Block[4,8] ✉

Both targeted therapies and immunotherapies provide benefit in resected Stage III melanoma. We hypothesized that the combination of targeted and immunotherapy given prior to therapeutic lymph node dissection (TLND) would be tolerable and drive robust pathologic responses. In NeoACTIVATE (NCT03554083), a Phase II trial, patients with clinically evident resectable Stage III melanoma received either 12 weeks of neoadjuvant vemurafenib, cobimetinib, and atezolizumab (*BRAF*-mutated, Cohort A, *n* = 15), or cobimetinib and atezolizumab (*BRAF*-wild-type, Cohort B, *n* = 15) followed by TLND and 24 weeks of adjuvant atezolizumab. Here, we report outcomes from the neoadjuvant portion of the trial. Based on intent to treat analysis, pathologic response (≤50% viable tumor) and major pathologic response (complete or near-complete, ≤10% viable tumor) were observed in 86.7% and 66.7% of *BRAF*-mutated and 53.3% and 33.3% of *BRAF*-wild-type patients, respectively (primary outcome); these exceeded pre-specified benchmarks of 50% and 30% for major pathologic response. Grade 3 and higher toxicities, primarily dermatologic, occurred in 63% during neoadjuvant treatment (secondary outcome). No surgical delays nor progression to regional unresectability occurred (secondary outcome). Peripheral blood CD8 + T$_{CM}$ cell expansion associated with favorable pathologic responses (exploratory outcome).

Immunotherapy with immune checkpoint inhibitors and targeted therapy with BRAF and MEK inhibition have transformed the treatment of melanoma. First tested in advanced disease, these treatments have led to durable responses and improved survival in patients with metastatic disease[1–4]. Both approaches have been applied as adjuvant therapy for completely resected high-risk Stage III melanoma following therapeutic lymph node dissection (TLND) with improvements in recurrence-free survival (RFS) for currently utilized regimens[5–7].

Despite these advances, a more effective therapeutic strategy is needed for the highest risk Stage III patients who present with operable disease yet have a substantial risk of relapse and death from melanoma[8].

Neoadjuvant systemic therapy for other surgically resectable cancers is used most frequently to downstage local and/or regional disease, test the efficacy of new therapeutic agents via pathologic response, and provide prognostic information and insight into mechanisms of resistance. Standard cytotoxic and targeted regimens largely have failed to show an overall survival advantage for neoadjuvant over adjuvant therapy[9]. More recently, pathologic response assessment after neoadjuvant therapy has been used to inform the need for or modification of adjuvant systemic therapies, which may in turn improve survival for those with resistant disease[10,11]. In contrast, and unique to neoadjuvant therapy with an immunotherapy backbone, there is a strong scientific basis for the hypothesis that treatment with the tumor in situ elicits more robust and durable anti-tumor immunity than the same treatment given in the adjuvant setting, suggesting that a neoadjuvant approach might improve cancer-specific and overall survival[12,13]. Unleashing the host immune response with the tumor in situ allows it to serve as an antigen depot for expansion and activation of tumor-specific T cells and other relevant immune cell subpopulations. This approach not only addresses clinically evident disease, but may also eradicate sub-clinical metastatic disease at other sites.

Several Phase II neoadjuvant melanoma trials addressing patients with high-risk stage III melanoma and oligometastatic resectable stage IV disease have been reported recently, including 6 trials evaluating preoperative immunotherapy and 2 evaluating preoperative targeted therapy[14–16]. These studies utilized varying approaches with neoadjuvant treatment ranging from 1 to 4 cycles over 3 to 12 weeks and adjuvant therapy ranging from 0 to 51 weeks for a total time on systemic treatment ranging from 6 to 54 weeks. Taken together, the data suggest that neoadjuvant immunotherapy doublets elicit a higher pathologic response rate than monotherapy, and a major pathologic response to immunotherapy has at least short-term durability. In parallel, data suggest that the combination of targeted and immunotherapy for metastatic *BRAF*-mutated (*BRAF*m) melanoma provides an improvement in progression-free survival relative to either targeted or immunotherapy alone[17–19]. For *BRAF*-wild-type (*BRAF*wt) melanoma, targeted therapies have not

been proven to provide benefit in the metastatic setting, but MEK inhibition has been demonstrated to synergize with anti-PD-1 in several preclinical models of cancer[20]; this approach showed promise in a Phase Ib clinical trial, although a subsequent Phase II trial (reported after our study was activated) did not substantiate this[21,22]. While these data suggest the possibility that combining targeted and immunotherapy may provide more benefit than either approach alone, prior to opening the present study there were no trials for high-risk surgically resectable stage III melanoma testing these combinations in the neoadjuvant setting.

Here, we report a Phase II trial in patients with high-risk clinically evident Stage III melanoma to evaluate the efficacy and tolerability of neoadjuvant combinatorial targeted therapy and immunotherapy. In this study, we test 12 weeks of neoadjuvant treatment with either vemurafenib, cobimetinib, and atezolizumab (in patients with *BRAF*m melanoma, Cohort A), or cobimetinib and atezolizumab (in patients with *BRAF*wt melanoma, Cohort B) followed by TLND and 24 weeks of adjuvant atezolizumab monotherapy. We hypothesize that the combination of targeted therapy and immunotherapy might be better than has been reported for either alone. We investigate the relationship between clinical outcomes and peripheral blood immune cell profiling via mass cytometry and flow cytometry, as well as expression of PD-L1 in tumor tissues and in plasma. The trial is designed such that the primary aim of the neoadjuvant component is pathologic response, while the primary aim of the adjuvant component is designated as RFS. Given recent emergence of data supporting pathologic response as predictive of RFS, we provide herein an interim report focusing on pathologic response to this combinatorial neoadjuvant approach. We also report on the secondary endpoint of adverse events occurring during neoadjuvant treatment and key blood and tissue biomarker translational analyses. RFS will be reported in a subsequent manuscript once these data are mature.

## Results

From June 22, 2018 to May 10, 2021, 30 patients (15 Cohort A, 15 Cohort B) with clinically evident Stage III surgically resectable melanoma were registered onto this trial and initiated therapy with patient flow as illustrated in the CONSORT diagram (Fig. 1). Over three quarters of patients presented with involvement of more than one lymph node, with the median clinical size of the largest affected node 3 cm. Patient, presentation, and tumor features are summarized in Table 1.

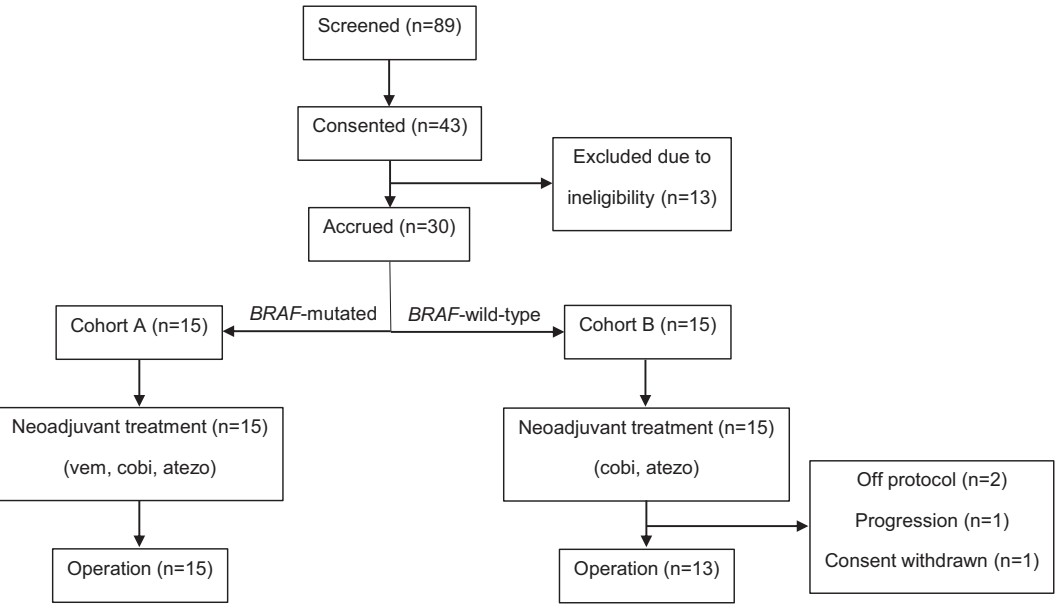

**Fig. 1 | CONSORT diagram for NeoACTIVATE (screening through operation).**

**Table 1 | Patient demographic and baseline characteristics**

| | Cohort | | Total |
|---|---|---|---|
| | A (N = 15) | B (N = 15) | (N = 30) |
| **Age, years** | | | |
| Median (IQR) | 56 (48–63) | 66 (59–73) | 59 (52–66) |
| Range | 22–66 | 44–82 | 22–82 |
| **Sex, n (%)** | | | |
| Female | 9 (60.0%) | 10 (66.7%) | 19 (63.3%) |
| Male | 6 (40.0%) | 5 (33.3%) | 11 (36.7%) |
| **ECOG Performance Status** | | | |
| 0 | 13 (86.7%) | 14 (93.3%) | 27 (90%) |
| 1 | 2 (13.3%) | 1 (6.7%) | 3 (10%) |
| **Presentation of Disease** | | | |
| De Novo | 10 (67.7%) | 7 (46.7%) | 17 (56.7%) |
| Recurrence | 5 (33.3%) | 8 (53.3%) | 13 (43.3%) |
| **Extent of Nodal Disease[a]** | | | |
| 1 involved lymph node | 4 (26.7%) | 3 (20%) | 7 (23.3%) |
| ≥2 involved lymph nodes | 11 (73.3%) | 12 (80%) | 23 (76.7%) |
| Diameter largest involved lymph node, cm, mean, median (range) | 3.3, 3 (1.2–6) | 4.2, 4 (1.4–11) | 3.8, 3 (1.2–11) |
| 1 involved nodal basin | 14 (93.3%) | 14 (93.3%) | 28 (93.3%) |
| 2 involved nodal basins | 1 (6.7%) | 1 (6.7%) | 2 (6.7%) |

[a]baseline from clinical and imaging assessment.

## Clinical activity, radiologic assessment, and pathologic response (primary outcome)

28 of 30 patients (15 Cohort A, 13 Cohort B) proceeded to TLND (as detailed in the protocol).

Eleven of the 15 Cohort A patients had measurable disease per RECIST 1.1 and had pre- and post-neoadjuvant radiographic assessment. Of these, 4 patients had a complete radiographic response, 6 had a partial response, and 1 had stable disease (Fig. 2). All 15 patients underwent per protocol operation. Ten of 15 patients 66.7% (10/15; 90% CI 42.3–85.8%) had a pathologic complete response (pCR); while 0, 3, and 2 patients had a near pathologic complete response (near-pCR, ≤10% viable tumor), pathologic partial response (pPR, 10.1–50% viable tumor), and pathologic non-response (pNR, >50% viable tumor), respectively (Fig. 3). Among 5 patients with persistent nodal disease at operation, 3 (60%) had one positive node. The details of pathologic response to treatment are summarized in Supplementary Table 1 and Supplementary Fig. 1. While a histologically mixed response was seen frequently, the dominant treatment response in the lymph node was necrosis/melanosis, with the frequency and extent of fibrosis/fibroinflammatory response lower among these patients. Pathologic and radiographic responses to treatment were not consistently correlated, as illustrated in Fig. 2B.

Nine of the 15 Cohort B patients had measurable disease per RECIST 1.1 and had pre- and post-neoadjuvant radiographic assessment. Of these, 4 had a partial radiographic response, 4 had stable disease, and 1 had progressive disease (Fig. 2). Thirteen of 15 patients proceeded to per protocol operation, of whom two patients had a pCR (13.3%, 2 of 15, per analysis of the intent-to-treat population, 90% CI 2.4–36.3%), while 3, 3, and 5 patients had a near-pCR, pPR, and pNR, respectively (Fig. 3). Among the 11 patients with residual nodal disease, 5 (45.5%) had disease limited to a single lymph node (Supplementary Table 1). The pathologic treatment response was again predominantly necrosis/melanosis with three notable exceptions in which histopathology assessment showed a pure fibrotic/fibroinflammatory response with no evidence of necrosis/melanosis (Supplementary Fig. 1). Similar to Cohort A, there were frequent inconsistencies between radiologic and pathologic assessments of response (Fig. 2).

## Treatment-related adverse events (secondary outcome)

The protocol-defined stopping criterion of development of Grade 4 non-hematologic toxicity in 2 or more of the first 6 patients or 30% or more patients thereafter was not met; thus accrual to the trial was not halted due to toxicity concerns.

All 15 Cohort A patients received neoadjuvant treatment. Thirteen patients completed all three planned cycles, while 2 patients completed two cycles. Eleven patients had a dose modification and 12 had a dose omission, both most often due to cutaneous toxicity. Eleven patients experienced at least one Grade 3 or 4 adverse event during neoadjuvant treatment (Supplementary Tables 2 and 3). The most frequent adverse events were rash (53.3%), hypertension (13.3%), and hyperglycemia (13.3%). No patient had surgical treatment delayed due to toxicity. One patient went off protocol after surgery due to Grade 3 pneumonitis.

All 15 Cohort B patients received neoadjuvant treatment. Twelve patients completed all three planned cycles, 1 patient completed two cycles, and 2 patients discontinued protocol participation after 1 cycle (one for progression with liver metastasis, the other per patient choice to withdraw to pursue alternative treatment). Seven patients had a dose modification, and 13 had a dose omission, both most often due to cutaneous toxicity or elevated liver enzymes. No patient had surgical treatment delayed as a result of toxicity. Eight patients experienced at least one Grade 3 or 4 adverse event during neoadjuvant treatment (Supplementary Tables 2 and 3). The most frequent adverse events were increased alanine aminotransferase (20%), hypertension (20%), and infection (13.3%).

## Surgical outcomes (secondary outcome)

As previously reported in a subset of these patients, technical aspects of operation were assessed in a structured fashion[21]. In the majority of patients there was no significant adverse effect of neoadjuvant combinatorial therapy on the technical conduct of the operation. Among the 28 patients completing operation per protocol, 9 of whom who had undergone prior nodal surgery of the same nodal basin, postoperative complications at 60 days were assessed per a pre-specified substudy protocol as well as through standard AE reporting. We observed complications attributable to operation in 8 patients (29%), all Grade 2, including seroma (6) and surgical site infection (2) (Supplementary Table 4). There were no wound dehiscences and no Grade 3+ complications.

## PD-L1 immunohistochemistry (exploratory outcome)

Quantitation of PD-L1 staining on tumor and immune cells from pre-treatment biopsies demonstrated no correlation between the frequency of PD-L1-positive tumor or immune cells and pathologic response to neoadjuvant therapy (Fig. 4A, B, Supplementary Figs. 2 and 3). Comparison of PD-L1 frequencies from pre-treatment biopsies and post-neoadjuvant surgical specimens showed, with a few exceptions, that pre- and post-neoadjuvant PD-L1 expression were similar (Fig. 4C).

## Soluble PD-L1 (exploratory outcome)

Soluble PD-L1 (sPD-L1) plasma concentrations from pre- and post-neoadjuvant samples showed a broad range of sPD-L1 levels. Pre-treatment sPD-L1 levels and the fold-change from pre- to post-neoadjuvant sPD-L1 were similar between those with pCR/near-pCR and those who had more residual disease (Fig. 4D, E, Supplementary Figs. 2 and 3). Correlation between tissue and soluble PD-L1 was poor (Fig. 4F, G, H; Supplementary Figs. 2 and 3).

## Mass cytometry (exploratory outcome)

In exploratory analyses, we assessed the frequencies of relevant immune cell populations in peripheral blood mononuclear cells (PBMCs) at 4 key timepoints (prior to treatment, after Cycle 1, after completion of neoadjuvant therapy, and after operation) using mass

A

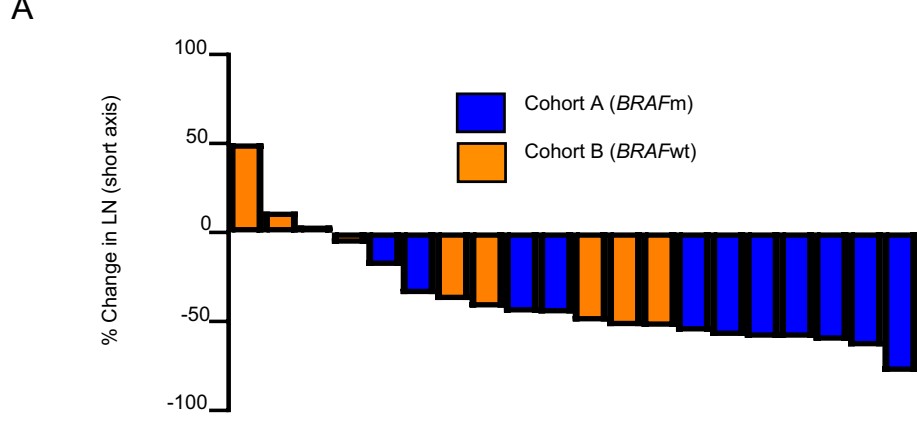

B

| Radiographic response | Pathologic response | | | | | | | |
| --- | --- | --- | --- | --- | --- | --- | --- | --- |
| | Cohort A (*BRAF* Mutant) (n=11) | | | | Cohort B (*BRAF* Wild-Type) (n=9) | | | |
| | pCR | Near-pCR | pPR | pNR | pCR | Near-pCR | pPR | pNR |
| CR | 3 | | 1 | | | | | |
| PR | 4 | | 1 | 1 | | 2 | 2 | 1 |
| SD | 1 | | | | | 1 | | 2 |
| PD | | | | | | | 1 | |

**Fig. 2 | Imaging Response to Neoadjuvant Treatment. A** Waterfall plot of radiographic responses to neoadjuvant treatment. The percent change in size of the largest lymph node from before to after neoadjuvant treatment is shown for all patients with measurable disease per RECIST 1.1 who completed adjuvant therapy on protocol and had post-neoadjuvant imaging assessment on protocol. **B** Comparison of Imaging and Pathologic Response.

cytometry and faceted by treatment cohort and response status (Fig. 5A). Frequencies of CD4 + T cells, CD8 + T cells, B cells, NK cells, and gamma-delta T cells are shown in Fig. 5B. Higher CD8 + T cell frequencies were seen in at multiple timepoints in patients with pCR or near-pCR ($\leq$10% viable tumor cells) relative to patients with more extensive residual disease. Since we saw an association between CD8 + T cells and pathologic response, we also assessed the frequencies of CD8+ naïve T cells, CD8+ central memory T cells ($T_{CM}$), CD8+ effector memory T cells ($T_{EM}$), and CD8+ effector memory T cells re-expressing CD45RA ($T_{EMRA}$). Notably, increased baseline $T_{CM}$ were observed in patients with favorable versus unfavorable pathologic responses (Fig. 5B). We also compared the intensity of PD-L1 staining on T cells over the course of neoadjuvant treatment and surgery. Relative to baseline, T cells from patients with *BRAF*m melanoma (Cohort A), who initiated anti-PD-L1 therapy with atezolizumab in Cycle 2, had roughly stable cell surface PD-L1 concentrations after Cycle 1 of neoadjuvant therapy and decreased cell surface PD-L1 after Cycle 3 (completion of neoadjuvant therapy) and Cycle 4 (surgery) (Fig. 5C). In contrast, T cells from *BRAF*wt patients, who received atezolizumab in Cycle 1, had decreased cell surface PD-L1 after Cycle 1, with cell surface PD-L1 levels remaining roughly stable thereafter. Differential abundance analysis between consecutive time points revealed a more extensive change of the immune cell profile in *BRAF*wt patients but not *BRAF*m patients (Supplementary Fig. 4A). The abundance of several CD4 + T cell subsets including naïve CD4 + T cells, CD4 + $T_{EM}$, and CD4 + $T_{CM}$ were found to significantly change over the treatment course in *BRAF*wt patients (FDR $\leq$ 0.2, Supplementary Fig. 4B). Specifically, the abundance of naïve CD4 + T cells decreased

after Cycle 1 relative to the baseline time point and recovered after completion of neoadjuvant treatment, with a concomitant increase in the abundance of CD4 + $T_{EM}$ and CD4 + $T_{CM}$.

**Flow cytometry (exploratory outcome)**
Using flow cytometry we quantitated three distinct blood immune cell populations: tumor-related T cells, effector cytotoxic T cells (CTLs), and pro-apoptotic T cells. Neither baseline frequencies of any of these populations nor the change in frequency from baseline to post-neoadjuvant treatment correlated with pathologic response (Supplementary Figs. 5−7).

## Discussion
Here we show that the combination of targeted therapy and immune checkpoint blockade is a promising neoadjuvant strategy for high-risk Stage III melanoma. To the best of our knowledge, this is the initial trial testing the neoadjuvant combination of BRAF and MEK inhibition with anti-PD-L1 immunotherapy in high risk surgically resectable Stage III *BRAF*m melanoma as well as testing neoadjuvant anti-PD-L1 immunotherapy with MEK inhibition for patients with *BRAF*wt disease. These regimens were efficacious, with a 70% rate of substantial pathologic response across the intention-to-treat population and a pathologic complete or near-complete response in 10/15 *BRAF*m and 5/15 *BRAF*wt patients. While this combinatorial regimen has not been tested previously, prior Phase 2 neoadjuvant trials have shown higher pathologic response rates to neoadjuvant targeted therapy (pCR/near-pCR rates 49−58%)[22–25] versus immunotherapy and for doublet immunotherapy combinations versus monotherapy (pCR rates 23−66.7% versus

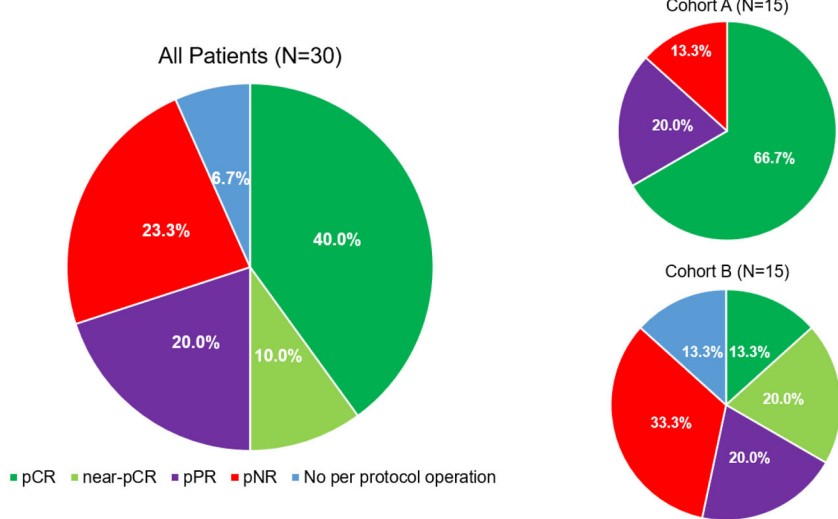

| Degree of Pathologic Response | Cohort A *BRAF*m (n = 15) | Cohort B *BRAF*wt (n = 15) | Total (n = 30) |
|---|---|---|---|
| | | | |
| Pathologic complete response (pCR) | 10 (66.7%) | 2 (13.3%) | 12 (40.0%) |
| Near- pathologic complete response (near-pCR) | 0 (0%) | 3 (20.0%) | 3 (10.0%) |
| Pathologic partial response (pPR) | 3 (20.0%) | 3 (20.0%) | 6 (20.0%) |
| Pathologic non-response (pNR) | 2 (13.3%) | 5 (33.3%) | 7 (23.3%) |
| No per protocol operation | 0 (0%) | 2 (13.3%) | 2 (6.7%) |

**Fig. 3 | Pathologic Response to Neoadjuvant Treatment.** The frequency and depth of pathologic response are shown in graphs and table for the entire intent-to-treat population and for each cohort.

21−29.6%)[15,26–30] among patients undergoing operation. The NeoTrio study, published in abstract form[29], similarly shows the highest pCR/ near-pCR rate (55%) in the combined therapy arm (dabrafenib, trametinib, and pembrolizumab). Our study adds to the growing body of data showing combinatorial neoadjuvant regimens drive higher pathologic response rates for resectable high-risk stage III melanoma.

The majority of patients in both arms of the study had a favorable response to neoadjuvant treatment as assessed by imaging and/or clinical examination prior to TLND. While one patient with clinical progression underwent imaging and operation after two rather than three cycles of neoadjuvant treatment, no patient was deemed unresectable nor failed to undergo operation due to regional progression. Interestingly, correlation between RECIST 1.1 radiographic responses to neoadjuvant treatment and pathologic responses was poor, with patients with complete and near-complete pathologic responses despite radiographic stable disease, and other cases with a pathologic partial response despite radiographic progression. These observations are in line with other neoadjuvant immunotherapy trials which have noted a significant discrepancy between the radiographic and pathologic responses to neoadjuvant treatment[15,28,30]. Based on these findings, we believe that while radiographic response assessment after neoadjuvant therapy is critical for identifying patients with distant disease progression who will not benefit from TLND, radiographic response should not serve as a surrogate endpoint for pathologic evaluation of the lymph node basin when assessing efficacy of neoadjuvant treatment.

Little data exist on the impact of neoadjuvant therapy on the technical aspects of operation and whether complication rates associated with TLND might be higher following various neoadjuvant regimens. Our acceptably low surgical complication rate provides

reassurance that these combinatorial regimens do not preclude safe operation. Despite the higher nodal disease burden in the present study, our surgical outcome data compare favorably with a recent report[30]. In that study, patients were treated with 2 cycles of neoadjuvant nivolumab 3 mg/kg and ipilimumab 1 mg/kg and 2/3 of patients enrolled had less extensive surgery consisting of limited nodal resection based on pathologic response. The investigators reported seroma rates of 33−52%, wound infection in 11−26% of patients and wound dehiscence in 3−23% of patients with the higher rates corresponding to patients treated with TLND. Additionally in that study 4 of 30 patients who had a TLND had the operation delayed due to ongoing irAEs. Multiple factors may contribute to these differences including differing neoadjuvant regimens, reporting definitions, disease burden, surgical technique and medical comorbidities.

While we observed excellent oncologic responses to neoadjuvant treatment, patients incurred significant toxicity during the neoadjuvant period, with 63% of patients experiencing at least one Grade 3 adverse event. Specifically, patients in Cohort A (vemurafenib, cobimetinib, and atezolizumab) experienced a high rate of Grade 3 cutaneous adverse events (53%); this is similar to the rate of grade 3−4 AEs (55%) seen in the NeoTrio trial testing 6 weeks of neoadjuvant dabrafenib, trametinib, and pembrolizumab for resectable stage III *BRAF*m melanoma[31]. Although toxicity from neoadjuvant therapy was generally reversible, did not delay surgery in any patient, and only precluded receipt of adjuvant atezolizumab in one patient; several patients required neoadjuvant treatment to be truncated, and the majority of Cohort A patients had a portion of neoadjuvant therapy omitted. While there were no new safety signals, neoadjuvant therapy with both targeted and immunotherapy requires careful monitoring of patients and frequent dose adjustments.

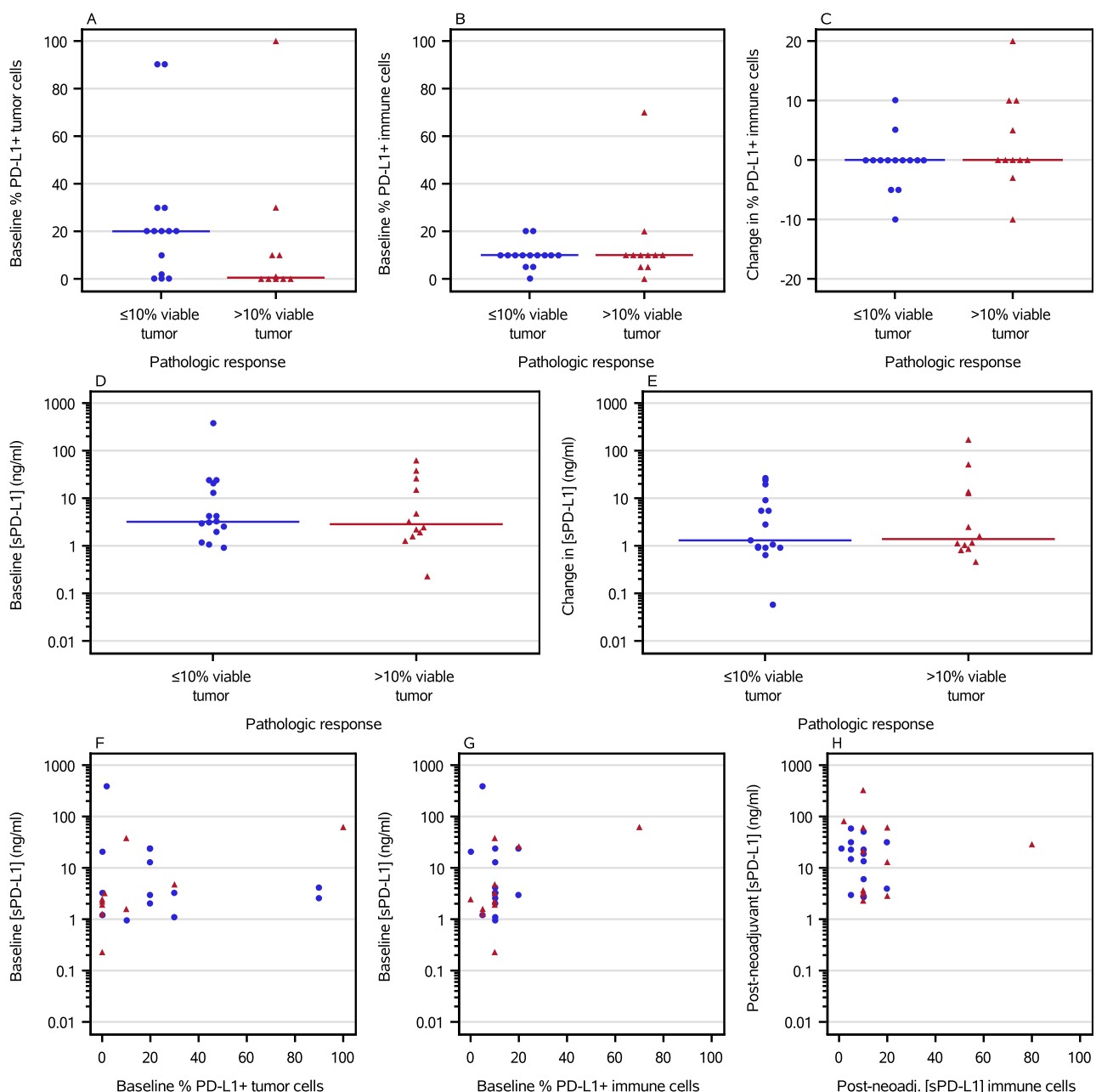

**Fig. 4 | PD-L1 IHC and sPD-L1 Analyses.** The percentage of tumor cells (**A**) and immune cells (**B**) with PD-L1 expression at baseline, and the change in percentage of immune cells with PD-L1 expression after neoadjuvant treatment (**C**) were quantitated. The plasma concentration of soluble PD-L1 at baseline (**D**) and the change in sPD-L1 concentration (**E**) were measured. Tissue PD-L1 staining on tumor (**F**) and immune cells (**G**, **H**) was compared at baseline (**F**, **G**) and after neoadjuvant therapy (**H**). Data are separated by treatment cohort in Supplementary Figs. 2 and 3. Blue circles are ≤10% viable tumor. Red triangles are >10% viable tumors. **A–E** lines are medians.

We observed a higher frequency of peripheral blood CD8 + T cells among those patients with a pCR or near-pCR at multiple timepoints, including at baseline, after one cycle of neoadjuvant therapy and after Cycle 4 (after operation); we also noted an increase in the frequency of CD8 + $T_{CM}$ cells at baseline in these patients. Like naïve T cells, CD8 + $T_{CM}$ cells home to secondary lymphoid organs including lymph nodes, where they interact with antigen-presenting cells and undergo antigen-specific clonal expansion; this suggests they may be of particular importance for eradication of disease present in lymph nodes. Immune checkpoint blockade in the setting of nodal disease should augment this response[32,33]. This subset of memory T cells is postulated to be particularly ideal in terms of its longevity combined with ability to exhibit effector functions upon reexposure to tumor antigens.

Additionally, data suggest continued exposure to antigen is not required for maintaining primed CD8 + $T_{CM}$ cells, implying that $T_{CM}$ can provide continued immunosurveillance even after disease eradication by systemic therapies and operation in the clinically affected lymph nodes[34]. We noted that in *BRAF*wt patients (but not *BRAF*m patients), CD4 + T cells decreased after Cycle 1 of neoadjuvant treatment and then recovered after Cycle 3. A possible explanation for the transient change in peripheral blood CD4 + T cell frequencies is that atezolizumab treatment leads to early mobilization of T helper cells into lymph node tissues (the site of the metastatic melanoma); however, this requires further investigation.

It was interesting to note that T cell surface PD-L1 expression decreased after Cycle 1 for patients in Cohort B, but the decrease was

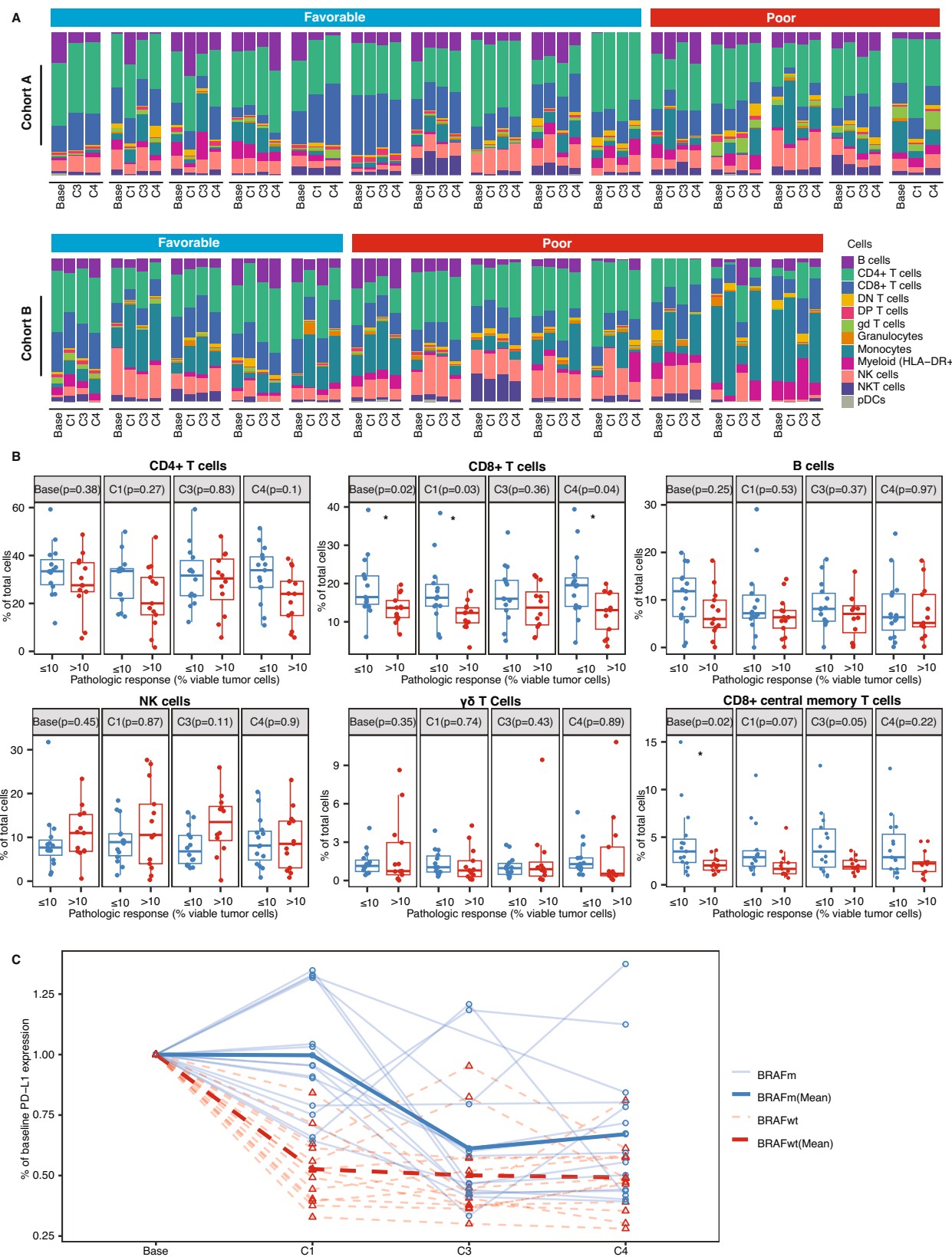

only seen after Cycle 3 for patients in Cohort A. We suspect that this is because patients in Cohort A did not receive atezolizumab until Cycle 2, whereas Cohort B patients received atezolizumab beginning in Cycle 1. Potential mechanisms for the decrease in cell surface PD-L1 expression include direct competition by atezolizumab for binding of the labeled antibody and atezolizumab-mediated internalization of PD-L1.

While we did not observe differences in pathologic response based on the degree of T cell surface PD-L1 downregulation, it is possible this may serve as biomarker of in vivo PD-L1 binding by atezolizumab.

Apart from the increase in CD8 + T cells and CD8 + $T_{CM}$ cells specifically, no other PBMC- or tissue-based candidate biomarkers that we examined appeared to be associated with pathologic response,

**Fig. 5 | Longitudinal analyses of peripheral blood immune cell populations.** Cellular subsets from peripheral blood samples were identified via CyTOF and quantitated at baseline, after Cycle 1, after completion of neoadjuvant treatment (Cycle 3), and after operation (Cycle 4). **A** Relative composition of immune cells at each timepoint in individual patients, faceted by treatment cohort and response status. **B** Frequencies of select lymphocyte subtypes as a percentage of total PBMCs, grouped by pathologic response status (favorable: pCR or near-pCR, ≤10% viable tumor cells; poor: pPR or pNR, >10% viable tumor cells). An asterisk (*)

denotes $p < 0.05$ (linear regression t-test, two sided adjusting treatmentarms). Jitters on the plot represent subjects. The lower and upper whiskers represent the range of data points within 1.5 times the interquartile range below or above the first and third quartiles respectively, while the box itself illustrates the middle 50% of the data, bounded by the first and third quartiles, with a median line indicating the data's median value. **C** Change from baseline intensity of PD-L1 expression on T cells in individual *BRAF*m and *BRAF*wt patients. Bold lines denote mean values for each cohort.

including tissue PD-L1, soluble PD-L1, or PBMC subpopulations based on flow cytometry or mass cytometry. It is possible this observation is due to the relatively small size of our study and the biomarker-driven treatment approach combining both targeted and immunotherapy. However, it is also possible that regional factors within the involved lymph node basin contribute substantially to pathologic response. Other than tissue PD-L1, regional biomarkers have not yet been assessed. Importantly, the primary rationale for treatment with neoadjuvant systemic therapy is not to eradicate nodal disease (which is resected anyway), but to prevent the development of overt distant metastases. It is quite possible that separate biomarkers may predict pathologic response in the involved lymph node basin(s) versus freedom from systemic recurrence of melanoma. We will assess the correlation of these candidate biomarkers with RFS once these data become mature.

Limitations of this study include the small sample size and lack of a comparative arm. Although all patients in the trial received immunotherapy, the high pathologic response rates we observed, particularly in Cohort A, may have been driven by targeted therapy and may or may not translate into a prolonged survival benefit. Additionally, these results are based on patients enrolled at two academic institutions; therefore, the findings may not be generalizable to patients treated in other settings.

With multiple studies of neoadjuvant therapy for high-risk Stage III and oligometastatic Stage IV melanoma being reported in the last several years[14,35], and with a recent report demonstrating superior event-free survival in patients receiving neoadjuvant and adjuvant versus adjuvant only pembrolizumab[16], neoadjuvant systemic treatment is emerging as a preferred approach for patients with clinically detected Stage III melanoma. However, several key questions regarding the optimal treatment approach remain unanswered. First, multiple combinatorial regimens have shown improved pathologic response outcomes relative to single agent pembrolizumab, including nivolumab/ipilimumab[28,30] and nivolumab-relatlimab[15]. Our study demonstrated a similarly high pathologic response rate in *BRAF*m patients; however, whether high pCR rates translate to increased survival and RFS will determine how the combination of vemurafenib, cobimetinib, and atezolizumab compares with immune checkpoint inhibitor combinations. Second, it has been proposed that pathologic response assessment after neoadjuvant treatment could be used to guide the extent of nodal surgery and the need for and nature of adjuvant systemic therapy; however, this has not yet been prospectively assessed. While our study did not alter surgical or systemic therapy based on pathologic response, this could be designed prospectively into future studies.

In summary, NeoACTIVATE is the first study to report outcomes from combination targeted and anti-PD-L1 immunotherapy in the neoadjuvant setting for patients with resectable high-risk Stage III melanoma. The rate of pathologic complete and near-complete responses in Cohort A compares favorably with other tested regimens. Cutaneous toxicity was significant, particularly in Cohort A, but did not delay surgical treatment. This approach appears promising, although comparison of RFS, distant metastasis-free survival and overall survival with other regimens is needed to determine the optimal neoadjuvant treatment strategy. Other correlative analyses are underway and we will report on these as well as survival outcomes when the clinical data mature.

## Methods
### Study design and participants
NeoACTIVATE, NCT03554083 (registered 5/29/2018), is an investigator-initiated, open label, two-arm, Phase II multi-center clinical trial carried out at Mayo Clinic (Rochester, MN) and the University of Minnesota (Minneapolis, MN), whose primary neoadjuvant aim is to examine the pathologic complete response (pCR) rate among patients with high-risk, resectable Stage III *BRAF*-mutant melanoma receiving vemurafenib-cobimetinib-atezolizumab (Cohort A) and among *BRAF*-wild-type patients receiving cobimetinib-atezolizumab (Cohort B). Eligible patients included patients aged 18 and older with clinically detected, recurrent or dual basin nodal metastatic melanoma confirmed histologically by pre-treatment needle biopsy. Patients who received prior systemic anti-cancer therapy or radiation were excluded, as were patients whose clinically evident disease was resected for diagnosis. Full details of the protocol, inclusion and exclusion criteria are provided in the protocol (see Supplementary Note). The study was conducted in accordance with Good Clinical Practice guidelines after approval of the institutional review boards at both Mayo Clinic and University of Minnesota and with oversight by an independent data safety monitoring board. Written informed consent was obtained from all participants prior to study registration, and re-consent was obtained after each update to the consent form due to changes in the investigational brochures for vemurafenib, cobimetinib, and atezolizumab. Remuneration for travel expenses was offered, but patients were not otherwise compensated. The study design and conduct complied with all relevant regulations regarding the use of human study participants and was conducted in accordance with the criteria set by the Declaration of Helsinki.

Thirty eligible patients enrolled from June 22, 2018 to May 10, 2021. All eligible patients who began treatment were included in the analysis of the primary endpoint. Patients who did not go on to surgery for any reason were considered not to have a pCR. The study was designed as a pilot study to gather preliminary anti-tumor activity, tolerability, and correlative data to inform further testing of this approach. A sample size of 15 patients per cohort was chosen so that the maximum half width of the 90% binomial confidence for the pCR rate would be +/− 21.2%.

### Interventions and assessments
All patients had pre-treatment staging including cross-sectional imaging and pre-treatment needle biopsy confirming melanoma and for *BRAF* testing. Neoadjuvant treatment had a cycle length of 28 days while adjuvant treatment had a cycle length of 21 days. Cohort A received neoadjuvant Cycle 1 treatment of 960 mg oral vemurafenib twice a day for days 1−22 and 720 mg oral vemurafenib twice a day for days 23−28, and 60 mg oral cobimetinib daily for days 1−21. Up to 2 more cycles of neoadjuvant treatment with 720 mg oral vemurafenib twice a day for days 1-28, 60 mg oral cobimetinib daily for days 1−21, and 840 mg intravenous (IV) atezolizumab on days 1 and 15 were administered. This dosing schedule followed that used in the IMspire150 phase III trial testing this combination in patients with metastatic melanoma[17]. Cohort B received up to 3 cycles of neoadjuvant treatment of 60 mg oral cobimetinib once daily for days 1−21 and 840 mg IV atezolizumab day 1 and 15. Patients were restaged with

cross-sectional imaging after Cycle 3 then proceeded to operation (TLND). Following operation patients in both Cohorts A and B received adjuvant 1200 mg IV atezolizumab Day 1 of a 21 day cycle for up to a maximum of 8 cycles. Adverse events during neoadjuvant and adjuvant therapy, as well as 60-day surgical complications were assessed using the CTCAE version 5.0 criteria. Specific events related to the operation were pre-specified in survey instruments and collected prospectively (Supplementary Table 4). Patients were followed by clinical examination and cross-sectional imaging every 12 weeks while on adjuvant therapy and for 2.5 years thereafter or until recurrence.

Radiographic response was ascertained according to RECIST version 1.1[36] for those patients with measurable disease. Pathologic response was assessed centrally by our study pathologist (TF) and reported according to the guidelines of the International Neoadjuvant Melanoma Consortium (INMC) for lymph node pathology assessments in neoadjuvant melanoma research trials[37]. Briefly, the total tumor bed area was assessed, and the proportion of viable tumor, if any, as a percentage of total tumor bed area was reported, along with the percentage of the tumor bed area exhibiting necrosis and/or fibrosis. Pre-specified benchmarks for major pathologic response were defined as 50% (Cohort A) and 30% (Cohort B) based on best estimates at time of study activation when little data existed for neoadjuvant treatment of Stage III melanoma.

Surgeon survey forms utilizing a structured assessment at baseline, day of operation, and 60 days after operation were completed. The baseline survey was designed to capture key impressions of disease burden at presentation including estimated number of involved lymph nodes, the size of the largest involved lymph node, and fixation to adjacent structures. The day of operation survey addressed the surgeon's impression of the degree of difficulty of the operation in comparison to the baseline assessment, as well as compared with the usual TLND of that basin. The 60 day survey focused on postoperative events including surgical site infections and other wound complications.

Formalin-fixed, paraffin-embedded tissue was obtained for research purposes at the time of pre-registration biopsy and at the time of TLND. Blood was obtained at the time of registration, after Cycles 1, 2, and 3 of neoadjuvant therapy, after Cycle 4 lymph node dissection. Blood was processed for viably cryopreserved peripheral blood mononuclear cells (PBMCs) and plasma per laboratory protocols[38].

### PD-L1 Immunohistochemistry
PD-L1 immunohistochemistry was performed (22C3 antibody) centrally and evaluated by the study pathologist (TF). Immunohistochemistry was performed on 4-μm thick formalin-fixed paraffin-embedded tissue sections using the following antibodies: PD-L1 22C3 (1:50 dilution; 22C3; Agilent Dako, Santa Clara, CA). Ultra Cell Conditioning solution (Ultra CC1) was used as a pretreatment step and OptiView DAB IHC Detection + OptiView AMP Detection System (Ventana Roche) was used for detection. Scoring was performed independently for tumor cells and immune cells on pre-treatment biopsy material and surgical tissues from operation following neoadjuvant treatment and assessed semi-quantitatively.

### Quantitation of sPD-L1 in plasma
Soluble PD-L1 was measured by ELISA per established protocols[39,40] from plasma specimens obtained at baseline, after Cycle 1, after Cycle 3, and after surgery. In brief, paired capture and detection antibodies to human PD-L1, mouse IgG2 monoclonal antibody clones H1A and B11 against extracellular human PD-L1, were utilized in a ELISA that used biotinylation and HRP-streptavidin detection system[41,42]. This assay is specific for sPD-L1 and does not exhibit cross reactivity to other B7 family molecules. Concentrations were

determined by optical density measurements according to a known standard curve of recombinant human PD-L1. ELISAs were performed by technologists who were blinded to the identity of the samples.

### Mass cytometry
Mass cytometry was performed on PBMCs using a panel of antibodies designed to discriminate different immune cell subsets and activation states focusing on T cells. The antibody panel used for this study is provided in Supplementary Table 6.

### Mass cytometry reagents
Culture medium (CM) was prepared with sterile RPMI-1640 media (Gibco) supplemented with 10% fetal bovine serum (Atlanta Biologicals) and PenStrep (Gibco). Benzonase Nuclease was purchased from Sigma-Aldrich. Maxpar® reagents including water, Cell Staining Buffer (CSB), Cell Acquisition Solution (CAS), Cell-ID Intercalator-Ir, Fix and Perm Buffer, Cell-ID™ 20-Plex Pd Barcoding Kit and EQ Four Element Calibration Beads were purchased from Fluidigm. Paraformaldehyde (PFA) was purchased from EM Sciences and 10X PBS pH 7.2 was purchased from Rockland. Antibodies used for cell surface labeling and phenotyping were purchased from Fluidigm. Custom conjugated antibodies were generated in-house through the Mayo Clinic Hybridoma Core using Maxpar X8 Ab labeling kits (Fluidigm) according to the manufacturer's protocol.

### Mass cytometry samples and processing
Cells were thawed and resuspended in CM containing 2.5 units/mL of Benzonase Nuclease (Sigma-Aldrich). After washing, cells were rested for 1 h in CM at 37 °C before staining. After resting $4 \times 10^6$ cells were resuspended in 1 mL of CSB. Each sample was incubated for 5 min with 0.5 mcm Cisplatin solution in PBS. Samples were then washed twice with CSB. An antibody cocktail of the entire phenotyping panel was prepared as a master mix prior to adding 50 mcL of cocktail to samples resuspended in 50 mcL of CSB. Samples were then incubated at room temperature for 45 min. After washing twice with CSB, samples were fixed with 2% PFA in PBS. After fixation and wash, samples were resuspended in 30 nM intercalation solution. Afterwards 30 mcL of unique barcoding reagent was added to each sample and incubated overnight at 4 °C. On the following morning cells were washed with PBS and pooled prior to resuspension in a 1:10 solution of calibration beads and CAS at a concentration of $0.5 \times 10^6$ cells/mL. Prior to data acquisition samples were filtered through a 35 mcm blue cap tube (Falcon).

### Mass cytometry data acquisition
Samples were loaded onto a Helios CyTOF® system (Fluidigm) using an attached autosampler and were acquired at a rate of 200−400 events per second. Data were collected as.FCS files using the Cytof software (Version 6.7.1014). After acquisition intra file signal drift was normalized to the acquired calibration bead signal using the CyTOF software.

### Mass cytometry statistical analysis
Normalized.FCS data were uploaded to the Astrolabe Cytometry Plateform (Astrolabe Diagnostics, Inc) where transformation, cleaning (doublets, debris), labeling, and unsupervised clustering were done. The resulting processed data were further analyzed by customized scripts. For differential abundance analysis of immune cell subsets, total sum scaling normalization and arcsine transformation were applied to the count data, and linear regression models were used to compare the abundance of cell subsets between response groups defined as favorable (≤ 10% viable tumor cells) versus unfavorable (>10% viable tumor cells) at different time points (baseline, C1, C3, C4), adjusting for treatment arms. Linear mixed-effects models were employed to evaluate changes in cell subset abundance across various time points (baseline compared to C1, C1 to C3, and C3

to C4) within the specific treatment arms of *BRAF*m and *BRAF*wt. The Benjamini-Hochberg (BH) procedure was applied for false discovery rate control, The PD-L1 marker intensity values across cells in each cell subset were averaged. The changes of the mean marker intensity from baseline to C1, C3, C4 were then calculated for the two treatment arms. All statistical analyses were performed using R software version 4.1.1.

### Flow cytometry of T cell subsets and activation states

Flow cytometry was performed on PBMCs from baseline, after Cycle 1, after Cycle 3, and after surgery using an antibody panel designed to identify tumor-related T cells, Effector cytotoxic T lymphocytes (CTLs), and pro-apoptotic T cells. To perform flow cytometry analysis of the phenotype and functional potential of immune cells, the following antibodies were used as per published protocols[43,44]: CD8-PE-Cy7 (clone RPA-T8, catalog 304006, BD Pharmingen), CD11a-APC (clone HI111, catalog 301212, BioLegend), PD-1 fluorescein isothiocyanate (FITC) (clone EH12.2H7, catalog 32990, BioLegend), CX3CR1-APC/Cy7 (clone 2A9-1, catalog 341616, BioLegend), Bim-PE (clone C34C5, catalog 12186S, Cell Signaling Technology), NKG7 monoclonal antibody (AG1490 Rb mAb 8H3/8K3)-FITC conjugated (Fusion antibodies with a contract with Dong lab). T cells were stained for surface markers before intracellular staining. Data was collected on a CytoFLEX LX (Beckman Coulter, Atlanta, GA) and analysis was performed with the R software version 4.1.1.

### Statistical considerations

The study was designed to assess the pathologic complete response rate within each cohort after the completion of neoadjuvant treatment. Pathologic complete response (pCR) rate was defined as the percentage of patients with no residual disease found in the surgical specimen of all patients who began neoadjuvant protocol treatment (intent-to-treat population). A patient is evaluable if registered, eligible, and started treatment. For analysis, 15 patients per cohort were to be enrolled to gather preliminary data to get point estimates for pCR. The confidence interval to be calculated using the binomial distribution. Pathologic response is sub-classified as the following: pathologic complete response (pCR, no viable tumor), pathologic near complete response (near-pCR, 0.1–10% viable tumor), partial pathologic response (pPR, >10.0–50% viable tumor) and minor or no response (pNR, >50% viable tumor). For analyses, patients were grouped as pCR and near-pCR versus pPR and pNR based on accepted definitions of major response to preoperative immune checkpoint blockade-containing regimens. Statistical analyses were performed using SAS software, version 9.4.

### Reporting summary

Further information on research design is available in the Nature Portfolio Reporting Summary linked to this article.

## Data availability

De-identified data are available with scientific approval of the study team under restricted access via email request to Matthew S. Block (block.matthew@mayo.edu). These data will be provided to scientific investigators for research purposes. Requests will be reviewed by the Institutional Review Board and subject to a Data Use Agreement. Details on acceptable methods and duration of data transfer will be determined by institutional policies based on which data are requested and for what research purposes. The remaining data can be found in the Article, Supplementary and Source Data files. Source data are provided with this paper.

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

## Acknowledgements

This investigator-sponsored work was funded by a Stand Up to Cancer Catalyst Award supported by Genentech, SU2C-AACR-CT10-17 (M.S.B.), and Mayo Clinic Center for Individualized Medicine Translational Research Award 92541640-2020 TRS#1 (T.J.H.) with additional support from the Department of Oncology, Mayo Clinic, Rochester, MN. The authors thank Kevin Pavelko (Mayo Clinic Immune Monitoring Core), Heidi Turner, Alisha Birgin, Renee Bradshaw, Simone Veum, Lynn Flickinger, Jill Schimke; Kenneth Hauer, and Tina Baxter for assistance with generation of mass cytometry data, development and conduct of the NeoACTIVATE clinical trial, and preparation of the manuscript.

## Author contributions

Conceptualization: TJH, MSB, Methodology and Analyses: TJH, GDN, TJF, EPG, JC, LAK, LY, RSD, CLE, CAS, SMH, VJS, HD, MSB, Trial coordination: TJH, ED, MSB, Investigation: TJH, TJF, EPG, JC, RRM, LAK, LY, ED, YY, SNM, AD, HNM, CLE, MAP, DLP, SSK, JH, SMH, MSB, Writing: TJH, GDN, TJF, EPG, JC, LY, CAS, VJS, HD, MSB, Final editorial review: all authors, Supervision: TJH, MSB, Funding: TJH, MSB.

## Competing interests

T.J.H.: Research support—Genentech, SkylineDX BV; R.R.M.: Research support—Bristol-Myers Squibb, GSK; L.A.K.: Advisory Board—Immunocore; R.S.D.: Research support—Bristol-Myers Squibb; S.N.M.: Research support—Bristol-Myers Squibb, Sorrento Therapeutics; Intellectual Property: Sorrento Therapeutics; A.D.: Honoraria—Intellisphere, Roche/Genentech; Advisory Board—TP Therapeutics, Guardant Health, AnHeart Therapeutics, ChromaCode; Clinical trial support—Syntrix Pharmaceuticals, Novartis, Merck, AnHeart Therapeutics, Sorrento Therapeutics, Guardant, Philogen, AstraZeneca; M.A.P.: Research funding—Intuitive Surgical; Honoraria—Kubtec Medical Imaging; D.L.P.: Advisory Board and Equity Interest—InSitu Biologics; Advisory Board—Medivis; M.S.B.: Research support—Alkermes, Bristol-Myers Squibb, Genentech, Merck, nFerence, Pharmacylclics, Regeneron, Sorrento Therapeutics, TILT Biotherapeutics, Transgene, Viewpoint Molecular Therapeutics; Consultant/Scientific Advisory Board—Sorrento Therapeutics, TILT Biotherapeutics, Viewpoint Molecular Therapeutics All other authors have no competing interests to declare.

## Additional information

[1]Division of Breast and Melanoma Surgical Oncology, Department of Surgery, Mayo Clinic, Rochester, MN, USA. [2]Department of Quantitative Health Sciences, Clinical Trials and Biostatistics, Mayo Clinic, Rochester, MN, USA. [3]Department of Laboratory Medicine and Pathology, Mayo Clinic, Rochester, MN, USA. [4]Department of Oncology, Mayo Clinic, Rochester, MN, USA. [5]Department of Quantitative Health Sciences, Computational Biology, Mayo Clinic, Rochester, MN, USA. [6]Division of Hematology, Oncology and Transplantation, University of Minnesota, Minneapolis, MN, USA. [7]Division of Hematology and Oncology, Department of Medicine, Mayo Clinic, Jacksonville, FL, USA. [8]Department of Immunology, Mayo Clinic, Rochester, MN, USA. [9]Department of Otolaryngology, Mayo Clinic, Rochester, MN, USA. [10]Department of Otolaryngology, University of Minnesota, Minneapolis, MN, USA. [11]Division of Surgical Oncology, University of Minnesota, Minneapolis, MN, USA. [12]Department of Urology, Mayo Clinic, Rochester, MN, USA. ✉e-mail: block.matthew@mayo.edu

