## [Peer Review File · Nature Communications]

Reviewers' Comments:

Reviewer #3:

Remarks to the Author:

My comments have been sufficiently addressed.

Reviewer #4:

Remarks to the Author:

A few minor suggestions in my relatively limited review:

1. The abstract does not detail the size of the study. I strongly recommend putting in the number of patients in each cohort in the abstract itself.
2. Combining MEK inhibition with immunotherapy in BRAFwt patients is somewhat unorthodox, though at the start of the study it may have been more accepted. I recommend adding additional references supporting this approach (as it stands, only one reference of a non-clinical study is cited).
3. The largest study describing MPR in neoadjuvant melanoma is S1801, which just reported a major pathological response in 50% (40% pCR) of patients receiving neoadjuvan pembrolizumab at ESMO 2023 by Patel et al. (The abstract can be found at [https://www.annalsofoncology.org/issue/S0923-7534\(23\)X0011-8?pageStart=49](https://www.annalsofoncology.org/issue/S0923-7534(23)X0011-8?pageStart=49) under LBA48). While not in the text of the abstract itself, in the presentation Dr. Patel also reported an 89% 24-month DFS among patients who achieved a path CR. While only an abstract, I think this would be important to cite given that it is now probably the benchmark for what immunotherapy alone would produce.

Response to Reviewers' Comments

****REVEWER #4****

1. The abstract does not detail the size of the study. I strongly recommend putting in the number of patients in each cohort in the abstract itself.

We agree and have added the number of patients in each cohort to the abstract.

2. Combining MEK inhibition with immunotherapy in BRAFwt patients is somewhat unorthodox, though at the start of the study it may have been more accepted. I recommend adding additional references supporting this approach (as it stands, only one reference of a non-clinical study is cited).

At the time the study was activated, there were no available clinical data on the combination of MEK inhibition and immunotherapy. However, since that time, several manuscripts have reported on this. We have included a sentence in the Introduction (Paragraph 3) referring to these data.

3. The largest study describing MPR in neoadjuvant melanoma is S1801, which just reported a major pathological response in 50% (40% pCR) of patients receiving neoadjuvant pembrolizumab at ESMO 2023 by Patel et al. (The abstract can be found at [https://www.annalsofoncology.org/issue/S0923-7534\(23\)X0011-8?pageStart=49](https://www.annalsofoncology.org/issue/S0923-7534(23)X0011-8?pageStart=49) under LBA48). While not in the text of the abstract itself, in the presentation Dr. Patel also reported an 89% 24-month DFS among patients who achieved a path CR. While only an abstract, I think this would be important to cite given that it is now probably the benchmark for what immunotherapy alone would produce.

We refer to the S1801 study in our manuscript (Reference 16; cited in Introduction, Paragraph 3 and Discussion, Paragraph 7). We did not report the 40% pCR rate noted in the abstract because the full manuscript describing the results of the trial reported a pathologic complete response rate of 21%.